# Pathogenicity Assessment of Colombian Strains of *Candida auris* in the *Galleria mellonella* Invertebrate Model

**DOI:** 10.3390/jof7060401

**Published:** 2021-05-21

**Authors:** Silvia Katherine Carvajal, Maira Alvarado, Yuli M. Rodríguez, Claudia M. Parra-Giraldo, Carmen Varón, Soraya E. Morales-López, José Y. Rodríguez, Beatriz L. Gómez, Patricia Escandón

**Affiliations:** 1Grupo de Microbiología, Instituto Nacional de Salud, Bogota 111321, Colombia; silvia.carvajal@javeriana.edu.co (S.K.C.); maira8822@gmail.com (M.A.); 2Grupo de Enfermedades Infecciosas, Departamento de Microbiología, Unidad de Proteómica y Micosis Humanas, Facultad de Ciencias, Pontificia Universidad Javeriana, Bogota 110231, Colombia; claudia.parra@javeriana.edu.co; 3Bacteriología y Laboratorio Clínico, Universidad Colegio Mayor de Cundinamarca, Bogotá 111311, Colombia; ymelissarodriguez@unicolmayor.edu.co; 4Hospital Infantil Napoleón Franco Pareja, Cartagena 130010, Colombia; cajuvame@yahoo.com.mx; 5Grupo CINBIOS, Departamento de Microbiología, Universidad Popular del Cesar, Valledupar 200004, Colombia; sorayaeugeniam@hotmail.com; 6Centro de Investigaciones Microbiológicas del Cesar, Valledupar 200002, Colombia; jyrodriguezq@gmail.com; 7Clínica Alta Complejidad, Valledupar 200002, Colombia; 8Clínica Laura Daniela, Valledupar 200002, Colombia; 9Instituto Cardiovascular del Cesar, Valledupar 200002, Colombia; 10Translational Microbiology and Emerging Diseases (MICROS), School of Medicine and Health Sciences, Universidad del Rosario, Bogota 111221, Colombia; beatriz.gomez@urosario.edu.co

**Keywords:** *Candida auris*, pathogenicity, *Galleria mellonella*, virulence, enzymatic activity, survival study

## Abstract

*Candida auris*, first described in 2009, is an opportunistic pathogenic yeast that causes nosocomial outbreaks around the world, with high mortality rates associated with therapeutic failure. In this study, we evaluated the pathogenicity of 107 isolates from two cities in Colombia, associated with fungemia or colonization processes; to achieve this, we used the *Galleria mellonella* invertebrate model to compare pathogenicity. Our results showed that less than half of the total isolates of *C. auris* presented a high pathogenicity compared to the reference strain SC5314, and most of those highly pathogenic strains were from colonization processes. We observed that there was formation of large aggregates of cells that cannot be disrupted easily, without statistically significant differences between the pathogenicity of the aggregated and non-aggregated strains. In addition, protease activity was observed in 100% of the *C. auris* strains; phospholipase and hemolysin activity were observed in 67.3 and 68.2% of the studied strains, respectively. In conclusion, these results highlight the utility of determining survival using *G. mellonella*, which allowed us to provide new information on the pathogenicity, enzymatic activity, and the relationship of the aggregated and non-aggregated phenotypes of *C. auris* in this model.

## 1. Introduction

*Candida* species are considered the most common fungi in hospital settings, with a notable incidence in the critically immunocompromised population, making them one of the most important etiological agents of invasive fungal infections [1]. Candidemia is the most common invasive fungal infection, associated with a high mortality of up to 40 and 60% in patients in intensive care units (ICU) [2]. The genus is composed of a heterogeneous group of organisms; one of these pathogens is *C. auris*, which has emerged as a serious public health problem, identified for the first time when recovered from the external auditory canal of a patient in Japan in 2009 [3,4]. This pathogen has rapidly emerged as an important nosocomial microorganism responsible for a significant number of invasive infections across the globe.

*C. auris* has been reported globally in many countries, such as South Korea, India, Pakistan, Kuwait, Israel, Oman, South Africa, Colombia, Venezuela, the United States, Canada, in Europe, including the United Kingdom, Norway, Germany, Spain, and the first case was already identified in Mexico [5]. The pathogen has been present in Colombia since at least 2015, with the first cases reported from the northern coastal region (Atlántico, Bolivar and Cesar) [6].

This yeast possesses virulence characteristics that include the production of enzymes such as proteases, phospholipases, and hemolysins, and the ability to colonize patients and their environments for weeks or months, which is reflected in its efficient transmission in health care centers. These factors play an important role in host tissue degradation, colonization, and resistance to oxidative stress, among other functions [7,8,9]. Another interesting characteristic of *C. auris* is that some isolates can grow in clumps, that is, with no release of daughter cells, resulting in complex aggregates that cannot be easily modified in vitro [10]; therefore, it could be thought that this characteristic may give them some resistance in the tissues and in the environment [11]. *C. auris* has been reported from a wide spectrum of clinical manifestations, ranging from colonization through deep-seated infections and candidemia [12,13]. Taken together, its multidrug resistance, rapid global emergence, and high mortality rates make this yeast a particularly problematic pathogen that has garnered considerable attention from the public, the medical community, and the basic research community [14].

The use of experimental models has been important for the study of the pathogenicity of this enigmatic yeast, helping to provide an approach to the physiopathology, virulence, and possible treatments. Murine models have been the most used over the years, although there are several bioethical, economic, and logistic complications that limit their research use [15]. The 3Rs principle (Replacement, Reduction, and Refinement) has been seen as a policy to guide the ethical use of animals in testing and encourages the development of alternative approaches to animal testing to provide better animal welfare and minimize the suffering. For this reason, different models that are easier to use have recently been chosen, and one of them is the invertebrate *Galleria mellonella*. In recent years, this experimental model has been crucial for understanding the pathogenicity of different *Candida* species, and relatively inexpensive compared to the murine model [16,17]. The advantage of using *G. mellonella* as a model for the study of pathogenicity in fungi is the ease of use, that is, the inoculation of the pathogen involves only a hind leg and a syringe, and they can be maintained at variable temperatures ranging from 25 to 37 °C. The innate immune response is an important component in the immune response to pathogenic infections and the reports show strong correlation between fungal pathogenicity results in larvae and mice [18,19].

In the present study, we aimed to compare the level of pathogenicity of 107 *C. auris* isolates from Valledupar and Cartagena, Colombia, by using the *G. mellonella* experimental model and determine the cell morphology (aggregated and non-aggregated cells) and the enzymatic activity of proteases, phospholipases, and hemolysins of the isolates.

## 2. Materials and Methods

### 2.1. Fungal Isolate and Preparation of Inoculum

One hundred seven isolates of *C. auris* were recovered from patients with invasive infections and colonization, in 7 medical institutions between 2016 and 2018 in Valledupar, *n* = 58 (isolates from invasive infections, *n* = 31; isolates from colonization processes, *n* = 27) and Cartagena *n* = 49 (isolates from invasive infections, *n* = 26; isolates from colonization processes, *n* = 23) (Table 1). Isolates were identified by matrix-assisted laser desorption ionization–time of flight (MALDI-TOF) and Polymerase Chain Reaction (PCR). All strains were maintained at −70 °C glycerol stocks. To perform the experiments, isolates were subcultured using Sabouraud Dextrose agar at 37 °C for 24–48 h. To determine the capacity of cells to form aggregates, vortex mixing was done for 1–2 min prior to microscopic examination.

To prepare the inoculum, the fungal cells were resuspended in sterile PBS and subsequently quantified using a Neubauer chamber; 10 μL of the suspension was placed on a glass slide and covered with a coverslip to be analyzed at 40X.

The ATCC SC5314 strain *C. albicans* was included in the evaluation of pathogenicity in *G. mellonella* and ATCC 90028 strain *C. albicans* was used as a positive control of enzymatic production, together with the *Cryptococcus neoformans* (H0058-I-743) and *Cryptococcus gattii* (H0058-I-860) moderate enzymatic activity strains.

### 2.2. Infection of Galleria Mellonella

Survival assays were performed in *G. mellonella*, using final (sixth) instar larvae, with an approximate weight of 300 mg, without melanization marks, stored at room temperature and in dark conditions. Before infection, the larvae were decontaminated with 0.1% sodium hypochlorite; controls of larvae without treatment (absolute-larval quality), inoculation control (control damage caused by inoculation) and larvae with only disinfection without being infected (disinfection) were used. Individual larvae were inoculated in the left rear proleg with 1 × 10^6^ yeast cells–larvae (final inoculum volume, 10 μL) using insulin syringe. At least 10 larvae were inoculated per isolate per experiment (triplicate experiment). After being inoculated, the larvae were placed at 37 °C in Petri dishes for dark incubation.

### 2.3. Enzymatic Activity 

Proteases: protease production was evaluated according to the protocol of Leone et al. [20]. Briefly, an inoculum from a Sabouraud agar culture of 72 h at 27 °C was adjusted to a McFarland scale 3 in 0.85% saline solution, seeded (5 μL) in the form of a drop in the center of a yeast carbon base (YCB) agar supplemented with bovine serum albumin and polypeptone, and subsequently incubated at 37 °C for 10 days. After this time, 5 mL of trichloroacetic acid was added for 1 h, the sample was washed with distilled water, and then 5 mL of Coomassie brilliant blue was added. Finally, the protease (Pz) index was evaluated by the halo formed around the yeast.

Phospholipases: phospholipase production was evaluated according to the protocol of Price et al. [21]. The same methodology described above for inoculum adjustment was used, and 5 μL was placed as a drop in the center of agar supplemented with egg yolk. The phospholipase (Pz) index value was determined through the ratio of the colony diameter plus the precipitation zone and the diameter of the colony.

Hemolysin: hemolytic activity was determined on Sabouraud agar with 3% glucose and 7% lamb blood. An inoculum of each isolate was prepared at a concentration of 1 × 10^8^ cells/mL, and 5 µL was seeded dropwise in the center of agar and incubated at 37 °C in 5% CO_2_ for 48 h. The hemolysis (Hz) index was evaluated by measuring the radius of the diameter of the colony with respect to the diameter of the translucent zone of the hemolysis (in mm) [22].

### 2.4. Mortality Study and Statistical Analysis

Death of the larvae was scored daily for 15 days. Kaplan–Meier survival plots were evaluated using the Log-rank (Mantel–Cox) test and *p*-values of ≤0.05 were used to indicate statistical significance. Statistics were performed using GraphPad Prism version 6.0 (GraphPad Software, San Diego, CA, USA). A 95% confidence interval was determined in all experiments. Another statistical analysis was performed using STATA 16 (STATA, College Station, TX, USA), via contingency tables and measures of association.

## 3. Results

### 3.1. Virulence of C. auris Isolates from Invasive and Colonization Processes

*G. mellonella* larvae showed different survival profiles depending on the fungal strain with which they were infected; 107 isolates of *C. auris* were studied, 57 isolates from fungemia and 50 associated with colonization processes. It was found that 35.5% (38/107) isolates of *C. auris* presented a high pathogenicity compared to the reference strain, when the death of the larvae was observed in an average of 5 days; of these isolates, 36.8% (14/38) were from invasive processes and 63.2% (24/38) were from colonization processes (Table 2; see Appendix A). There was a statistically significant difference between the virulence and type of process with *p* = 0.011, suggesting that colonization isolates were more virulent than those from invasive processes. Eight out of 107 strains were randomly chosen to represent the survival graph to show their statistical analysis (Figure 1A,B; see also Appendix A).

### 3.2. Virulence Comparison of Aggregated and Non-Aggregated Cells of C. auris

In the survival curves, it was observed that 58% (22/38) of the isolates considered virulent and comparable with the SC5314 presented a non-aggregating morphology. The total larval death for the non-aggregating virulent isolates had a mean between 2 and 3 days, while that of virulent aggregates was between 4 and 5 days, but the difference was not statistically significant (*p* = 0.124) (Figure 2) (Table 3).

### 3.3. More Than Half of the Isolates with High Pathogenicity Were Obtained from Cartagena

Of 107 isolates in the study, 49 and 58 were from Cartagena and Valledupar, respectively. It was observed that of the 38 isolates that pathogenically behaved similarly to the reference strain SC5314, 21 were strains from Cartagena and 17 from Valledupar, but the difference was not statistically significant (*p* = 0.145).

### 3.4. Enzymatic Activity

All isolates in the study presented some protease activity: 36.4% (39/107) weak activity, 58% (62/107) moderate activity, and 5.6% (6/107) strong protease activity. No or weak phospholipase activity was observed: only 2.8% (3/107) had moderate to strong activity. Weak activity was presented in 37.4% (40/107) of the isolates, 26.2% (28/107) moderate activity, and only 4.7% (5/107) strong hemolytic activity. Overall, our results showed that 100% of the isolates were positive for the production of proteases, 67.3% (72/107) for the production of phospholipases, and 68.2% (73/107) for hemolytic activity.

Of the 38 isolates with a high pathogenicity, 71% (27/38) had a moderate to high protease activity; no isolates with moderate or strong phospholipase activity were observed, and 39.5% (15/38) had moderate to strong hemolytic activity (Figure 3; Appendix A).

## 4. Discussion

*Candida auris* has been recently identified as an enigmatic pathogen, with very particular characteristics that have been associated with infection and outbreaks in healthcare settings in six continents [5,14].

The *G. mellonella* model has been previously proved to be reliable for the study of pathogenicity of fungal strains according to their virulence or for preliminary drug testing [23]. In this work, we used *G. mellonella* as an in vivo model to evaluate the pathogenicity of 107 *C. auris* strains, comparing it with *C. albicans* reference strain (SC5314).

First, we observed that 38% of the isolates in the study had a pathogenicity comparable to *C. albicans*, which is currently accepted as the most pathogenic member of the genus [24] and many of these isolates were associated with colonization processes. Colonization isolates can persist in the environment and on the patient’s skin [25]; as there are still no unified criteria to define a case of *C. auris* infection, it is complicated to characterize colonization rates, screening techniques, and clinical relevance of colonization for the development of invasive infection. Colonization by *C. auris* has been detected at multiple body sites, including nose, groin, axilla, and rectum, and it has been isolated over 3 months or more after initial detection. The contact time needed for acquisition of *C. auris* has been suggested to be short hours, and invasive infections have been described within 48 h of admission to ICU [26,27]. Between 2018 and 2019, 349 cases of *C. auris* were reported in the European Union, 257 (73.6%) were colonizations, and 84 (24.1%) were bloodstream infections [28]. Therefore, the importance of an early and correct identification of patients colonized with *C. auris* and a rigorous nosocomial control to prevent the spread, and a possible invasive infection especially in severely immunocompromised patients [29]. In the review by Fasciana T. et al., they mention some recommendations for the control and prevention of the spread of this microorganism by colonization, knowing that there is no established decolonization protocol: i. once a case of *C. auris* is detected, screening is recommended to detect patients in close contact; ii. immediate notification to public health authorities; iii. education of healthcare workers on the importance and impact of *C. auris*; iv. notification of the state of colonization/infection by *C. auris* in transferred patients [30].

Some *C. auris* strains exhibit an aggregative phenotype, that is, they are able to survive forming cellular aggregates embedded in the same cell wall due to an incomplete budding [24]. Although aggregated cells are generally more tolerant and resistant to various environments than their non-aggregating counterparts, there are some studies that report that aggregated cells show less virulence compared to non-aggregated cells in *G. mellonella* [24,31]. However, in our study, we did not find significant differences between larvae mortality produced by aggregated and non-aggregated *C. auris* strains (*p* = 0.124), in contrast to other studies that found higher mortality in groups of larvae infected with non-aggregated strains. Though *C. auris* virulence is comparable to *C. albicans* at the concentrations used, Sherry et al. showed, at lower concentrations (1 × 10^5^ and 5 × 10^4^ cells/larvae), some non-aggregating cells are more virulent than *C. albicans* in *G. mellonella* [31]. Our result is in agreement with a study by Romera et al. in 2020 where they reported that they did not find differences in larvae mortality induced by aggregated and non-aggregated strains of *C. auris* [32].

More studies are needed to relate these results to the biology of this fungus and to clarify the association with the outcome of the patient. In Colombia, there is a study by Muñoz et al. reported in 2020, in which virulence of *C. auris* and *C. haemulonii* complex species was compared, using *G. mellonella* larvae to evaluate the survival rate, fungal burden, histopathology and phagocytosis index, survival using BALB/c mice and analyze the fungal capacity to form biofilm over an inert surface. The results showed that the larvae and mice survival rate was lower when infected with *C. auris* strains than when infected with the *C. haemulonii* species complex [17].

Identifying virulence factors in *Candida* strains can help to understand their adhesion, invasion, and infection process. Enzymes play an important role in the pathogenicity of yeasts because they are involved in the destruction of host tissues and provide access to nutrients for yeasts, facilitating the colonization process [33].

In the present study, we evaluated the production of three types of extracellular enzymes in *C. auris:* proteases, phospholipases, and hemolysins, which as mentioned are crucial virulence factors for the pathogenesis of *Candida* spp. Our results reflect substantial production of proteases (100%) and more than half of the isolates had phospholipase activity (67.3%). The production of proteinases and phospholipases has been recognized as relevant for *Candida* spp. contributing to the pathogenicity of these species and helping in the adhesion and invasion of host cells [34]. It was observed that more than 60% of the isolates had hemolytic activity (68.2%); this enzyme can degrade hemoglobin by lysis of erythrocytes to release iron, which is used for growth and metabolism in the case of systemic infections [35]. Junqueira et al. evaluated the enzymatic activity of 64 isolates of *Candida* spp.; among the 31 non-*albicans* species, 21% had proteinase activity and 39% had phospholipase activity. Previous studies reported that *C. albicans* produces high levels of proteases and phospholipases, whereas non-*albicans* species commonly have lower production of these enzymes [36]. Larkin et al. reported in their study that *C. auris* phospholipase production depended on the strain and the production of this enzyme was detected in 37.5% of the isolates analyzed. They mentioned that, in general, the *C. auris* strains that produced phospholipase tended to have weak phospholipase activity, correlating with our study [9]. Despite the high positivity in the production of enzymes, it was observed that most of the isolates producing enzymes were isolates with low pathogenicity in the invertebrate model *G. mellonella*. Our results differ with some reports such as that of Rossoni et al., who reported that for strains of *Candida* spp. with higher proteinase and phospholipase activity, the pathogenicity in *G. mellonella* was higher [37]. The reason for this difference is not clear; in the study by Rossoni et al., different *Candida* species were used and did not include strains of *C. auris*. *Candida* spp. pathogenesis has been reported to be directly related to the quantity and potency of these enzymes, as well as to the source and site of isolation [38]. The production of these enzymes and the association with virulence in *C. auris* should be further explored.

Colombia has been one of the countries that has reported cases of infection or colonization by *C. auris* since 2016 [39,40]. The virulence of the strains differed according to the region; although this difference was not so marked, the strains from Cartagena showed more virulence than those from Valledupar. It would be helpful to analyze the clade of these strains together with other strains from different regions of the country. It would also be interesting to compare these results with hemocyte count, fungal load, and the histopathological analysis of the infected larvae.

## 5. Conclusions

*C. auris* presents virulence factors of considerable interest, which make this pathogen a potential adapter to different environmental and cellular processes. In our study, less than half of the strains had virulence comparable to *C. albicans.* Of the virulent strains, many were predominantly from colonization processes, suggesting that these colonizing strains have the capacity to be more pathogenic for patients. In contrast, the absence or presence of cell aggregation processes showed no clear relation to virulence in this study. 

Our study contributes results from in experiments with the invertebrate model *G. mellonella* that clarified the association of pathogenicity with other characteristics of isolates of *C. auris.* These findings could serve as a basis for future experiments in mammalian models that would provide further information pertinent to *C. auris* infection in human, on the virulence factors and the relationship between the phenotypes of strain and virulence.

## Figures and Tables

**Figure 1 jof-07-00401-f001:**
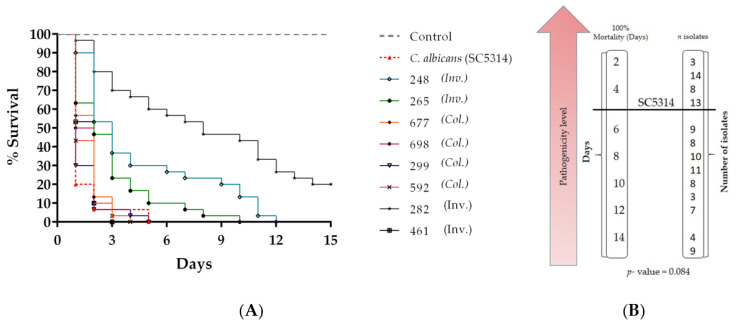
Survival curves of *G. mellonella* infected with 1 × 10^6^ cells/larvae of *C. auris* and *C. albicans* (**A**) Kaplan–Meier plots of *G. mellonella* survival after injection of *C. auris* from invasive (*Inv.*) and colonization (*Col*.) processes compared to *C. albicans*, which is characterized as high virulence. (**B**) Number of isolates that caused 100% mortality of the larvae in the different days 1–15 of post-infection follow-up compared with strain SC5314.

**Figure 2 jof-07-00401-f002:**
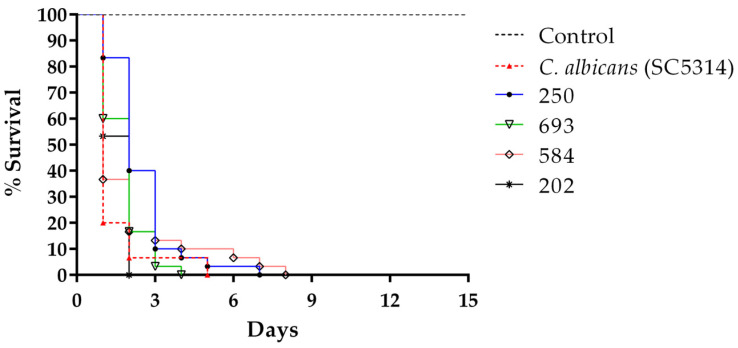
Virulence of aggregate-forming and non-aggregate strains of *C. auris* compared to *C. albicans* SC5314 in *G. mellonella* larvae at 37 °C. Kaplan–Meier plots of *G. mellonella* survival after injection with 10^6^ cells/larva of *Candida albicans* (red line), non-aggregating *C. auris* strains (blue and black line), and aggregate-forming *C. auris* strains (pink and green line) are shown. Experiments were performed in triplicate with 10 larvae per strain in each experiment. No larval killing was observed in control larvae injected with an equivalent volume of PBS (black arrowed lines).

**Figure 3 jof-07-00401-f003:**
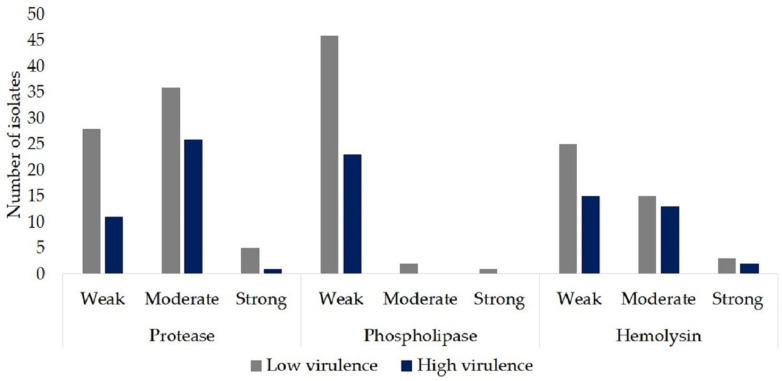
Comparison of the virulence of *C. auris* isolates from the present study versus the enzymatic activity.

**Table 1 jof-07-00401-t001:** Strains used in this study are shown. Number of isolates and type of morphology of *C. auris* recovered from invasive and colonization processes in Colombian patients (*p* = 0.814).

	*n* Strains	Morphology		Process	
		Aggregates	Non-Aggregates	Invasive	Colonization
Valledupar	58	22	36	31	27
Cartagena	49	13	36	26	23
**Total**	**107**	35	72	57	50

**Table 2 jof-07-00401-t002:** Strains of *C. auris* with high and low virulence, exhibiting a statistically association with processes (*p* = 0.011).

Virulence	Process		Total
	Invasive	Colonization	
High	14	24	**38**
Low	43	26	**69**
Total	**57**	**50**	**107**

**Table 3 jof-07-00401-t003:** Strains of *C. auris* with high and low virulence tested with the type of cell morphology, the difference was not statistically significant (*p* = 0.124).

Virulence	Morphology		Total
	Aggregates	Non-Aggregates	
High	16	22	**38**
Low	19	50	**69**
Total	**35**	**72**	**107**

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
