# Peer review of "Pathogenicity Assessment of Colombian Strains of Candida auris in the Galleria mellonella Invertebrate Model"

_jof, 2021, doi:10.3390/jof7060401_

Round 1
Reviewer 1 Report
The paper described the pathogenicity assessment of 107 isolates of Candida auras from 7 centers from 2 cities in Colombia. The authors used the Glleria mellonella model to analyze the pathogenicity factors. The manuscript is very well written, the methods are adequate and the authors group has expertise in this field.
1) In the introduction, the authors mentioned complications of using murine models. Could they provide examples relevant to this field of research?
2) Could they comment the practical advantage of using this model to test pathogenicity?
3) To improve the discussion of diagnostics related to colonization capacity of Candida auras, I may suggest to include specific papers addressing this in the reference, such as doi: 10.3390/antibiotics9110778
4) In the discussion, the authors mentioned differences between their findings and those by Rossoni et al. Could the authors comment on the reasons for this difference?
Author Response
We thank the reviewer for the very important comments made to the manuscript.
Reviewer 1
Changes:
- In the introduction, the authors mentioned complications of using murine models. Could they provide examples relevant to this field of research?
Line 78, 79, 80 and 81: The 3Rs principle (Replacement, Reduction and Refinement) has been seen as a policy to guide the ethical use of animals in testing and encourages the development of alternative approaches to animal testing to provide better animal welfare and minimize the suffering.
Line 87, 88, 89 and 90: The innate immune response is an important component in the immune response to pathogenic infections and the reports show strong correlation between fungal pathogenicity results in larvae and mice.
2) Could they comment the practical advantage of using this model to test pathogenicity?
Line 84, 85, 86 and 87: The implementation of G. mellonella as a model for the study of pathogenicity in fungi is the ease of use, that is, the inoculation of the pathogen involves only a hind leg and a syringe and they can be maintained at variable temperatures ranging from 25 to 37 °C.
3) To improve the discussion of diagnostics related to colonization capacity of Candida auris, I may suggest to include specific papers addressing this in the reference, such as doi: 10.3390/antibiotics9110778
Line between 266 and 277: Between 2018 and 2019, 349 cases of C. auris were reported in the European Union, 257 (73.6%) were colonizations and 84 (24.1%) were bloodstream infections [28]. There is the importance of an early and correct identification of patients colonized with C. auris and a rigorous nosocomial control to prevent the spread, colonization and a possible invasive infection especially in severely immunocompromised patients [29]. In the review by Fasciana T. et al., They mention some recommendations for the control and prevention of the spread of this microorganism by colonization, knowing that there is no established decolonize tion protocol: i. Once a case of C. auris is detected, screening is recommended to detect patients in close contact; ii. Immediate notification to public health authorities; iii. Education of healthcare workers on the importance and impact of C. auris; iv. Notification of the state of colonization / infection by C. auris in transferred patients [30].
4) In the discussion, the authors mentioned differences between their findings and those by Rossoni et al. Could the authors comment on the reasons for this difference?
Line 332, 333 and 334: The reason for this difference is not clear; in the study by Rossoni et al. different Candida species were used and did not include strains of C. auris.
Reviewer 2 Report
The authors aimed to assess the pathogenicity of Colombian isolates of Candida auris using a Galleria mellonella infection model and determining the driving factors behind this pathogenicity by measuring enzymatic activity.
I have no major concerns over this manuscript and the study has been well thought out and executed. I just have some minor points, mainly concerning the results section:
For me, the title is a bit too long. Perhaps removing ‘isolated from invasive infections and colonization processes’ would make the title sound more appealing. This is personal preference and not essential.
Figure 1 is difficult to distinguish strains. Making this figure larger would improve readability.
Despite a number of enzymatic assays being carried out, there are no figures to accompany this. Either figures/images or tables should be provided (either in the main manuscript or as supplementary) to provide an easier to read way of showing proportions of strains possessing enzymatic properties.
Statistical significance should be included in tables.
C. auris virulence is comparable to C. albicans at the concentrations used. However, Sherry et al (2017) showed, at lower concentrations (1x105 cells/larvae), some non-aggregating cells are more virulent than C. albicans in G. mellonella. This could be addressed in the discussion.
Author Response
We thank the reviewer for the very important comments made to the manuscript
Reviewer 2
Changes:
- For me, the title is a bit too long. Perhaps removing ‘isolated from invasive infections and colonization processes’ would make the title sound more appealing. This is personal preference and not essential.
Tittle was changed as suggested. Line 3: Pathogenicity assessment of Colombian strains of Candida auris in the Galleria mellonella invertebrate model
- Figure 1 is difficult to distinguish strains. Making this figure larger would improve readability.
Size of figures of survival was adjusted: 5.98- 5.96 and 11.91 cm (Figure 1 and 2).
- Despite a number of enzymatic assays being carried out, there are no figures to accompany this. Either figures/images or tables should be provided (either in the main manuscript or as supplementary) to provide an easier to read way of showing proportions of strains possessing enzymatic properties.
Supplementary material annex 3 is being included for consideration.
- Statistical significance should be included in tables.
Every table has included the statistical significance in the manuscript. Added statistical significance (line 233).
- auris virulence is comparable to C. albicans at the concentrations used. However, Sherry et al (2017) showed, at lower concentrations (1x105cells/larvae), some non-aggregating cells are more virulent than C. albicans in G. mellonella. This could be addressed in the discussion.
Line 286, 287 and 288: Though C. auris virulence is comparable to C. albicans at the concentrations used, Sherry et al. showed, at lower concentrations (1 x 105 and 5 x 104 cells/larvae), some non-aggregating cells are more virulent than C. albicans in G. mellonella.